# Gold Nanoclusters Grown on MoS_2_ Nanosheets by Pulsed Laser Deposition: An Enhanced Hydrogen Evolution Reaction

**DOI:** 10.3390/molecules26247503

**Published:** 2021-12-11

**Authors:** Yuting Jing, Ruijing Wang, Qiang Wang, Xuefeng Wang

**Affiliations:** 1Shanghai Key Lab of Chemical Assessment and Sustainability, School of Chemical Science and Engineering, Tongji University, Shanghai 200092, China; jyt1215@163.com (Y.J.); 1510586@tongji.edu.cn (R.W.); 2State Key Laboratory of Coal Conversion, Institute of Coal Chemistry, Chinese Academy of Sciences, Taiyuan 030001, China

**Keywords:** molybdenum disulfide, Au nanoparticles, pulse laser deposition, hydrogen evolution reaction, density functional theory

## Abstract

Au nanoparticles were decorated on a 2H MoS_2_ surface to form an Au/MoS_2_ composite by pulse laser deposition. Improved HER activity of Au/MoS_2_ is evidenced by a positively shifted overpotential (−77 mV) at a current density of −10 mA cm^−2^ compared with pure MoS_2_ nanosheets. Experimental evidence shows that the interface between Au and MoS_2_ provides more sites to combine protons to form an active H atom. The density functional theory calculations found that new Au active sites on the Au and MoS_2_ interface with improved conductivity of the whole system are essential for enhancing HER activity of Au/MoS_2_.

## 1. Introduction

Electrocatalytic water splitting is a promising technique to produce hydrogen, which is a green energy source; however, highly active platinum-based catalysts are expensive, which severely impedes the development of electrocatalytic hydrogen evolution reaction (HER) techniques [1]. During previous decades, alternatives, such as metal alloys [2], transition metal carbides [3], nitrides [4], borides [5], phosphides [6] and chalcogenides [7], have been extensively explored to replace Pt-based HER catalysts. Among them, molybdenum disulfide (MoS_2_) receives unusual attention because of its low cost, earth-abundance and intrinsic HER activity [8]. Researchers have proven that the edge site of 2H MoS_2_ is the active site for electrochemical HER; but limited edge sites make natural 2H MoS_2_ show lower catalytic activity [9]. In addition, semi-conductive 2H MoS_2_ has poor electron conduction in HER processes, impeding its expression of activity.

To overcome the drawbacks above, strategies to improve HER activity of MoS_2_ have been implemented by exposing more edges or adhering MoS_2_ onto conductive substrates [10,11]. For example, various metal nanoparticles such as Pt and Pd have been incorporated into MoS_2_ nanosheets, which effectively improves MoS_2_ activity [12]. Remarkably, MoS_2_ decorated with Au nanoparticles is especially effective for HER due to strong Au-S interaction and the excellent stability of Au particles [13,14]. Au nanoparticles were selectively decorated on the edges and line defects of MoS_2_ basal planes by a spontaneous redox reaction, and the Au nanoparticles improved the charge transport of MoS_2_, thus enhancing the HER catalytic efficiency [15]. Zhang et al. prepared ultra-small MoS_2_-Au nanohybrids by a solvothermal method, which showed enhanced HER catalytic activity. A synergistic effect between Au and MoS_2_ promoted the activity of edge sites and enhanced the conductivity [16]. Zhao et al. constructed a composite of Au_25_ clustered with thiolate and selenolate ligands on MoS_2_ nanosheets, which exhibited enhanced HER activity. The dual interfacial effect, namely interfacial electronic interactions between gold nanoclusters and MoS_2_ nanosheets, and the interface between metal core and surface ligands, improved the charge transfer and electronic interactions [17]. Obviously, the interface of catalysts plays a vital role in tuning the catalytic performance.

In this work, ligand-free Au nanoparticles were decorated on a 2H MoS_2_ surface to form a composite (Au/MoS_2_) by pulse laser deposition (PLD). The 2H MoS_2_ nanosheets possess a large, exposed surface which provides ideal support for landing Au nanoparticles. The optimized Au/MoS_2_ composite with Au (111) nanoparticles on a 2H MoS_2_ basal plane showed improved HER activity. Based on the density functional theory (DFT) calculation, it was found that Au nanoparticles modulate the electronic structure of MoS_2_ and improve electron conductivity of the whole system. Significantly, the interfacial effect of the Au/MoS_2_ interface is critical for enhancing HER activity of Au/MoS_2_.

## 2. Results and Discussion

The morphology of prepared samples was studied by field emission scanning electron microscopy (FESEM) and transmission electron microscopy (TEM) as shown in Figure 1. Tiny Au nanoparticles were deposited on Ti foil as a control sample by PLD in Figure 1a. Large, ultrathin MoS_2_ nanosheets were prepared by a hydrothermal process as shown in Figure 1b. The obtained 3D morphology of MoS_2_ nanosheets gives several structural advantages: the open flower-like structure provides a large surface area to load more Au particles; the initial laminated nanosheets produced the active edge sites as much as possible. Figure 1c shows the Au/MoS_2_ with a Au deposition time of 5 min on MoS_2_ nanosheets, which shows a very uniform distribution of Au particles (~3 nm). Figure 1d shows a TEM image of MoS_2_ nanosheets, in which MoS_2_ shows the obvious layer structure with a layer space of 0.65 nm. Figure 1e shows a TEM image of Au nanoparticles on the MoS_2_ nanosheets and Figure 1f is the HRTEM image of the Au anchored on the MoS_2_ nanosheets, which shows lattice spacing of 0.243 nm corresponding to the (111) plane of Au and 0.23 nm to the (103) of MoS_2_. This elaborate interfacial structure benefits from the pulsed laser process. To be specific, when the Au plasma plume interacts with the S atom layer on the MoS_2_ basal plane, the high supersaturation of Au vapor starts to nucleate. Subsequently, larger Au islands are formed through recrystallization between deposition pulses. Due to the lowest surface energy of the Au (111) plane, it has become the final prior orientation on the MoS_2_ surface [18].

The crystal structure of the prepared nanocomposites was investigated by XRD measurement as shown in Figure 2a. The XRD spectra of the MoS_2_ nanosheets reflect the peaks at 14.2° (002) and 33.6° (100), which is indexed to 2H-MoS_2_ (JCPDS: 37-1492). In the case of Au/MoS_2_, there is not any additional peak for the Au crystals, showing the nano feature of the Au particles. Raman spectroscopy is further used to expose the structural difference between MoS_2_ and Au/MoS_2_ as shown in Figure 2b. The pristine MoS_2_ nanosheets show two peaks at 379 cm^−1^ and 408 cm^−1^, which correspond to active E^1^_2g_ and A_1g_ modes of Mo-S bonding. The E^1^_2g_ mode represents the in-plane vibrations between the Mo layer and two S layers, and the A_1g_ mode corresponds to the out-of-plane lattice vibration with S atoms moving in the opposite direction [19]. After introducing the Au nanoparticles into the MoS_2_ nanosheets, the A_1g_ mode of MoS_2_ shows a blue shift because of the strengthened vertical vibration of S atoms by the interaction between Au and MoS_2_. The large dielectric constant of Au could enhance the screening of the electron-electron interactions and weaken the planar interionic interactions, leading to a very slightly red shift of the E^1^_2g_ peak of Au/MoS_2,_ which softens the Mo-S phonon mode of MoS_2_ and decreases the intensity of Mo-S chemical bonds [20].

In Figure 3, XPS spectra were used to characterize the bonding between Au and MoS_2_. Two peaks at 229.2 eV and 232.3 eV for MoS_2_ and Au/MoS_2_ are attributed to the Mo 3d_5/2_ and Mo 3d_3/2_, respectively. The S 2p_3/2_ and S 2p_1/2_ peaks of MoS_2_ show the binding energy of 162.0 eV and 163.3 eV in Figure 3b [13]. The significant change of Au/MoS_2_ is that the doublet peaks of S 2p become fuzzy and a new doublet at the lower binding energy in Figure 3c reveals the formation of an Au-S bond between Au and MoS_2_ [16]. The peaks in Figure 3d at 84.3 eV and 88.0 eV correspond to the Au 4f_7/2_ and Au 4f_5/2_ for Au/MoS_2_, respectively. Compared with the pure Au element at 84.0 eV, the slightly higher Au 4f_7/2_ binding energy means a charge transfer between Au and S, which is in keeping with the S 2p spectrum of Au/MoS_2_ and further certifies the formation of an Au-S bond [21]. All this certified that the Au nanoparticles anchor more strongly with MoS_2_, and the electron transfer between them would benefit hydrogen adsorption in the electrocatalytic process [22].

To evaluate the HER performance of the prepared catalysts, a three-electrode system was used in 0.5 M H_2_SO_4_ conditions. In Figure 4a, to reach a current density of 10 mA cm^−2^ for H_2_ evolution, Au/Ti shows an overpotential of 352 mV and MoS_2_ shows 313 mV. In sharp contrast, Au/MoS_2_ gives a lower overpotential of 236 mV than Au and MoS_2_ at the same catalytic current density, indicating the advantage of the interaction between MoS_2_ and Au. All the Tafel plots show a value smaller than the 120 mV dec^−1^ but larger than the 30 mV dec^−1^ in Figure 4b, suggesting the Volmer–Heyrovsky mechanism, where the combination between electron and proton to bond another hydrogen atom to form the hydrogen molecule is the rate-determining step. The exchange current density (*j*_0_) is an inherent feature of HER catalysts and is usually obtained by an extrapolation method [23]. The *j*_0_ of Au, MoS_2_ and Au/MoS_2_ is 0.62, 5.98 and 28.92 μA cm^−2^, respectively. It is obvious that Au/MoS_2_ has an excellent *j*_0_ compared to other catalysts, showing that the Au/MoS_2_ surface has superior intrinsic activity for electrochemical HER, because the addition of Au on the MoS_2_ surface improves the structure of the initial MoS_2_.

For further certification of the catalysts, ac impedance spectroscopy was conducted using frequencies from 10 kHz to 0.1 Hz. As shown in Figure 4c, Au/MoS_2_ exhibits smaller charge transfer resistance than Au and MoS_2_ according to the smaller diameter of the semi-circle in the middle frequency, suggesting very fluent electron transfer in hydrogen formation. Besides HER activity, Au/MoS_2_ also gives excellent HER stability during the HER process in Figure 4d. By testing cyclic voltammetry ranging from 0 to −0.425 V (vs. RHE) in 0.5 M H_2_SO_4_ solution, there was no distinct change of catalytic current after 1000 cycles, showing good stability of Au/MoS_2_.

To further evaluate the influence of Au on HER performance in the Au/MoS_2_ system, the content of Au on the MoS_2_ surface was adjusted by controlling the depositing time. As seen in Figure 5, the morphology of Au/MoS_2_ had an obvious change with the increase of deposition time. Specifically, the Au nanoparticles were hardly observed in the sample (1 min) Au/MoS_2_ and (3 min) Au/MoS_2_ (Figure 5a,b), but the indistinct trail of Au appeared in the sample with 5 min Au loading. Further extending the deposition time to 10 min, the MoS_2_ nanosheets were covered with Au film as shown in Figure 5c. The analysis of element distribution of 5 min Au/MoS_2_ in Figure 5f from the domain of Figure 5e shows uniform distribution of Au on the MoS_2_ surface. Figure 5g,h show the element percentage of Au in the corresponding area, and the atom percentage of Au is 1.53 at. % for (5 min) Au/MoS_2_, which is lower than that of 10 min Au sample (3.35 at. %) but higher than that of 3 min (1.11 at. %) and 1 min (0.87 at. %), showing the controlled content of Au by PLD progress.

The electrochemical HER activity of different Au deposition time samples is shown in Figure 6a, in which the activity of Au/MoS_2_ had an obvious change as Au content increased. Specifically, the best performance was when deposition time was 5 min, but when extending deposition time to 10 min the activity of the sample decreased. The value of the Tafel slope of Au/MoS_2_ in these conditions still showed the Volmer–Heyrovsky mechanism in Figure 6b. Figure 6c shows the EIS results, the charge transfer resistance of samples decreased as the Au amount on MoS_2_ increased, suggesting a positive effect of Au on MoS_2_ activity. However, larger resistance appears with more Au deposition (10 min) on the substrate, which shows a saturation of Au distribution. Coverage of the whole MoS_2_ surface results in the disappearance of the interface between Au and MoS_2_, thus losing the corresponding active sites and decreasing the activity. The electrochemically active surface area (ECSA) derived from the double-layer capacitance (C_dl_) is usually utilized to reflect the exposure of active sites of catalysts, and the C_dl_ is calculated by conducting the cyclic voltammetry curves in different scan rates in Figure 6d. The 5 min Au/MoS_2_ catalyst had a larger C_dl_ of 11.1 mF cm^−2^ than the other films, showing there was greater exposure of active sites. Decreased overpotential, lower charge-transfer resistance and many more exposed active sites in the 5 min Au/MoS_2_ sample imply the interfacial importance between Au and MoS_2_ for HER.

The interaction between the Au nanoparticle and the 2H MoS_2_ basal plane were calculated by DFT calculation. For comparison the perfect 2H MoS_2_ and Au/MoS_2_ nanoribbon (Figure 7a) were calculated. The Δ*G*_H*_ values of MoS_2_ (2.21 eV) and Au/MoS_2_ (0.19 eV on the Au atoms) are shown in Figure 7b. The lower Δ*G*_H*_ of the Au/MoS_2_ models explains the reason for improved HER performance by the Au/MoS_2_ catalyst, in which the combination of new Au sites on the interface and the initial edge sites on MoS_2_ provided more active sites for HER. Figure 7c shows the band structure of MoS_2_, which had a wide bandgap of 1.62 eV, implying its semiconductor nature [24]. After combining Au with the MoS_2_ basal plane, the bandgap of Au/MoS_2_ gets smaller (0.25 eV) as shown in Figure 7d, showing that the improved Δ*G*_H*_ is from the change of electronic structure in the catalyst. A narrower bandgap also indicates that Au had a strong interaction with the MoS_2_ basal plane, thereby accelerating electron conductivity in Au/MoS_2_ [25]. Definitely, the more d-orbital electrons of Au than that of S made some electronic states close to the Fermi level, suggesting more occupied states in the valence band of Au/MoS_2_. Figure 7e shows the total density of states (TDOS) of MoS_2_ and Au/MoS_2_. Compared with the pristine MoS_2_, the TDOS of Au/MoS_2_ has an obvious left shift, indicating an increase in the number of electrons for the MoS_2_ [25]. The partial density of states (PDOS) of Au/MoS_2_ is shown in Figure 7f, it was found that the 5d orbital of Au atom formed a hybrid with the 3p orbital of S atoms and the 4d orbital of adjacent Mo atoms, which promoted rapid charge transfer on the systems [26]. Finally, the lower Δ*G*_H*_, a smaller bandgap and sufficient interface of Au/MoS_2_ made the number of active sites increase and the whole conductivity improve, thus enhancing the HER activity of Au/MoS_2_.

It should be mentioned that more active Au sites were located on the interface between Au and MoS_2_ rather than the top Au atom of the Au nanoribbons, implying the importance of the Au/MoS_2_ interfacial structure. The function of the interface could also be proven by our Au/MoS_2_ samples with different Au loading. The 1 min and 3 min Au/MoS_2_ samples showed relatively low activity because the Au nanoparticles were too few to provide sufficient interfacial sites to adsorb H atoms, while the 5 min Au/MoS_2_ catalyst exhibited high activity (Figure 6). Complete coverage of Au (10 min) on the MoS_2_ surface blocked the active Au sites, leading to a reduction in HER activity. Both experiments and theoretical calculations confirmed that the interface between Au and MoS_2_ provided more sites to combine protons to form the active H atom. In addition, the interfacial structure was also beneficial in promoting the Heyrovsky reaction in the Volmer–Heyrovsky mechanism, because the proton from the solution and the electron from the electrode could easily combine with the H atom adsorbing on the interfacial Au sites [27].

The study of Au/MoS_2_ addresses that tunable Au on MoS_2_ nanosheets is a potential system for examining the potential HER mechanism between Au and MoS_2_. Firstly, Au/MoS_2_ has improved activity compared with pure MoS_2_; secondly, the Au/MoS_2_ interface could be adjusted by changing the depositing time, which further adjusts HER activity. Finally, the interfacial structure between Au and MoS_2_ is important for advancing HER activity, as proved by a lower bandgap of Au/MoS_2_ and more active interfacial Au sites with a lower Δ*G*_H*_ based our DFT calculations.

## 3. Materials and Methods

### 3.1. Preparation of MoS_2_ Nanosheets on Ti Foil

The MoS_2_ nanosheets were synthesized by a facile hydrothermal reaction. At first, the Ti foil ((0.5 mm., Sinopharm Chemical Reagent Co., Ltd.)) was treated by acetone and deionized water under ultraphonic conditions. Then, the MoS_2_ nanosheets were grown on the substrate. A 30 mL mixed water solution containing NaMoO_4_∙2H_2_O (120 mg)(A.R., Sinopharm Chemical Reagent Co., Ltd.) and CH_4_N_2_S (190 mg) (A.R., Sinopharm Chemical Reagent Co., Ltd.) was used as the reactant to prepare the MoS_2_ in an oven at 200 °C for 20 h [12]. After that, the obtained sample was taken out and rinsed with water and ethanol repeatedly before drying in air at 60 °C.

### 3.2. Deposition of Au Nanoparticles on MoS_2_ Nanosheets

The Au nanoparticles on the MoS_2_ surface were deposited by pulsed laser deposition (PLD). Before the deposition process, a stainless steel chamber was pumped to a base pressure of 1 × 10^−4^ Pa by using a molecular turbo pump. The Au target (purity > 99.99%, Alfa Aesar) was ablated by a focused Nd: YAG laser beam with a wavelength of 1064 nm at 10 Hz at 25 °C. The laser energy set in the laser device was 330 mJ cm^−2^, and the spot area in the deposition was about 1 mm^2^ on the target. The distance between the Au target and the substrates was set to 3 cm. The pressure in the deposition was maintained at 1.2 × 10^−3^ Pa. Compared with the initial pressure of 1 × 10^−4^ Pa, the Au plasma pressure was about 1.1 × 10^−4^ Pa. The quantity of Au on the MoS_2_ substrates was controlled by different deposition times, which were 1 min, 3 min, 5 min and 10 min. For better identification of these samples, the labels of Au-1/MoS_2_, Au-3/MoS_2_, Au-5/MoS_2_ and Au-10/MoS_2_ were used in the description to distinguish the deposition time of the Au. The most effective Au/MoS_2_ sample was equal to Au-5 on the MoS_2_ nanosheets. Each sample had three parallel contrasts as shown in Appendix A.

### 3.3. Material Characterization

The microscopic morphology and element ingredients of all synthesized samples were characterized by field emission scanning electron microscopy (FESEM; Hitachi S-4800, Tokyo, Japan), energy dispersive X-ray spectroscopy (EDS, Tokyo, Japan), and high-resolution transmission electron microscopy (TEM; JEOL, JEM-2100, Tokyo, Japan). X-ray photoelectron spectra (XPS, Kratos Axis Ultra DLD, America), Raman spectroscopy (514 nm laser, Rainshaw Invia, UK) and X-ray diffraction (XRD, Bruker Focus D8 with Cu Ka radiation, Germany) were utilized to characterize the crystal structure, composition and valence state of the samples.

### 3.4. Electrochemical Measurements

The electrochemical properties were evaluated in a three-electrode system by a CHI760D electrochemical workstation (Chenhua Co., Shanghai, China). Au/MoS_2_/Ti served as a working electrode, a graphitic rod (diameter: 3 mm) as a counter electrode and a saturated calomel electrode (SCE) as a reference electrode in 0.5 M H_2_SO_4_ electrolyte at room temperature. The Pt/C reference catalyst was prepared by dispersing 5 mg of commercial 10 wt% Pt/C in 0.5 mL of ethanol with 10 μL of 5 wt% Nafion solution with the dip-coating process on a glassy-carbon electrode. A reversible hydrogen electrode (RHE): *E(RHE)* = *E(Hg/Hg_*2*_Cl_*2*_)* + 0.059 pH + 0.242 V was used as a standard to evaluate the performance of different catalysts. The EIS were conducted over a frequency domain from 10 kHz to 0.1 Hz at an amplitude of 5 mV.

### 3.5. Calculation Method

To understand the active sites of Au/MoS_2_ in HER, we carried out first-principle calculations on MoS_2_ with a different model, which was based on density functional theory (DFT) employing projector-augmented wave potentials (PAW) [28]. All calculations were carried out with Perdew–Burke–Ernzerhof (PBE) in the Vienna Ab initio Simulation Package (VASP) [29]. A plane wave cutoff of 500 eV was used and the Brillouin zone of the surface calculation was 3 × 3 × 1 Monkhorst–Pack mesh. The convergence criteria was 1 × 10^−5^ eV for energy, and total forces on each atom were 0.02 eV/Å in ionic relaxation. A quasi-one-dimensional Au nanoribbon with a thickness of two layers and a width of two atoms on planar MoS_2_ was used [30].

The Gibbs free energy for hydrogen adsorption Δ*G*_H*_ was used to theoretically recognize the HER activity of the catalysts, in which appropriate hydrogen binding on the catalysts was usually reflected by an optimal value of 0 eV. It is defined as [31]
(1)$$\Delta G_{H^{*}}=\Delta E_{H}+\Delta ZPE-T\Delta S_{H}$$
where Δ*E*_H_ is the binding energies of the nanoribbon and hydrogen atom, defined as
(2)$$\Delta E_{H}=E_{slab-H}-E_{slab}-\frac{1}{2}E_{H_{2}}$$
where *E*_slab-H_ is the total energy for the catalyst which absorbed a hydrogen atom, *E*_slab_ is the total energy for the intrinsic catalyst and *E*_(H2)_ is the energy of a gas phase hydrogen molecule. Δ*E*_ZPE_, the difference in zero-point energy between the adsorbed and gas phase, is calculated to be 0.04 eV for H/MoS_2_. Δ*S*_H_, the entropy difference between the adsorbed state and the gas phase standard state, can be approximately expressed as Δ*S*_H_ ≈ −1/2S^0^_H2_ due to the negligible vibrational entropy in the adsorbed state, where S^0^_H2_ is the entropy of the gas phase H_2_ at standard conditions. So −*T*Δ*S*_H_ ≈ −1/2S^0^_H2_ = 0.20 eV [32]. Hence,
Δ*G*_H*_ = Δ*E*_H_ + 0.24(3)

## 4. Conclusions

In conclusion, Au nanoparticles were decorated on a 2H MoS_2_ surface to form a composite (Au/MoS_2_) by pulse laser deposition. The optimized Au/MoS_2_ composite with Au (111) nanoparticles on a 2H MoS_2_ basal plane shows improved HER activity. The DFT calculation shows that the addition of new Au sites optimizes hydrogen adsorption free energy in Au/MoS_2_. In addition, the interfacial structure between Au and MoS_2_ provides favorable conditions to promote formation of hydrogen. At the same time, improved conductivity of the whole electrode ensures fluent electron transfer. Finally, Au/MoS_2_ shows a lower overpotential of 236 mV compared with 313 mV of MoS_2_ at the current density of 10 mA cm^−2^.

## Figures and Tables

**Figure 1 molecules-26-07503-f001:**
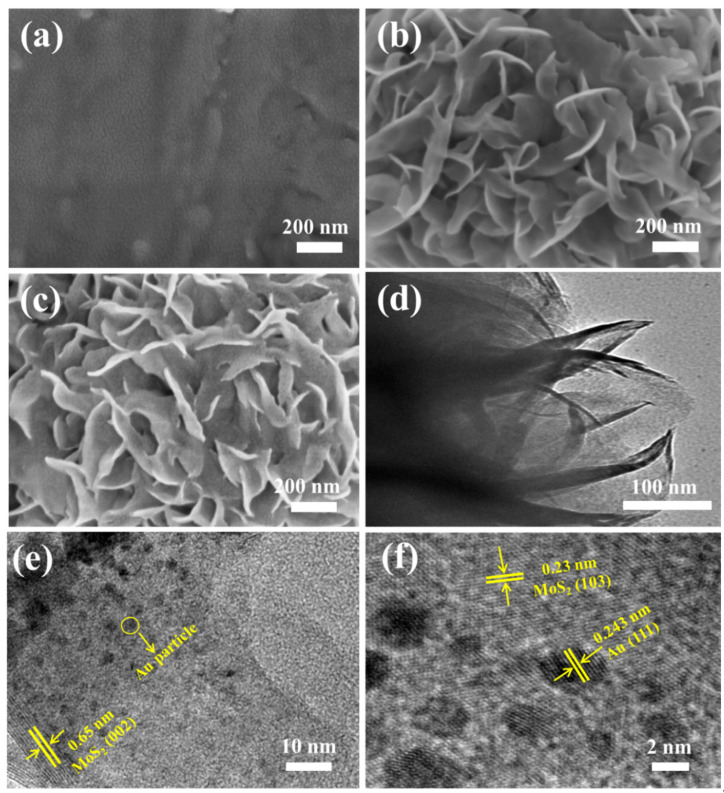
(**a**) FESEM images of the Au/Ti, (**b**) MoS_2_/Ti and (**c**) Au/MoS_2_/Ti; (**d**) TEM images of the MoS_2_, (**e**) Au/MoS_2_ and (**f**) HRTEM image of Au/MoS_2_.

**Figure 2 molecules-26-07503-f002:**
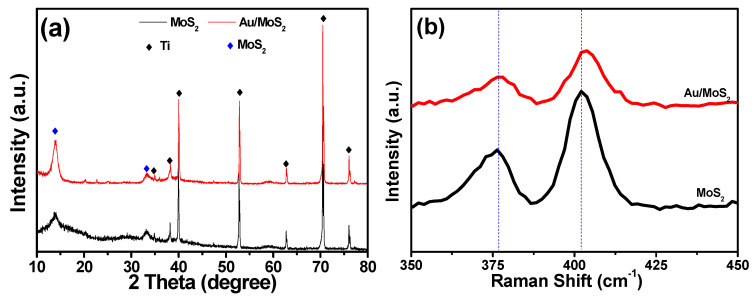
(**a**) X-ray diffraction patterns and (**b**) Raman spectra of MoS_2_ and Au/MoS_2_, respectively.

**Figure 3 molecules-26-07503-f003:**
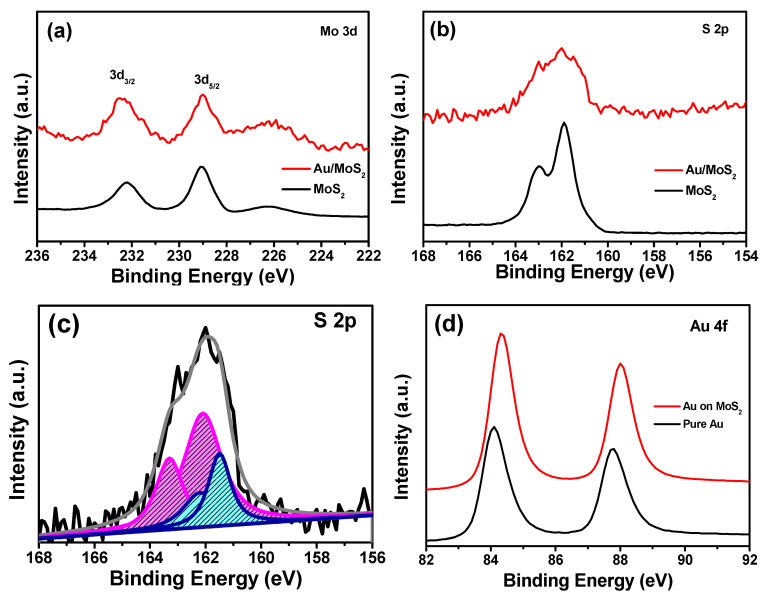
The XPS analysis of MoS_2_ and Au/MoS_2_: (**a**) Mo 3d, (**b**) S 2p, (**c**) S 2p and (**d**) Au of Au/MoS_2_.

**Figure 4 molecules-26-07503-f004:**
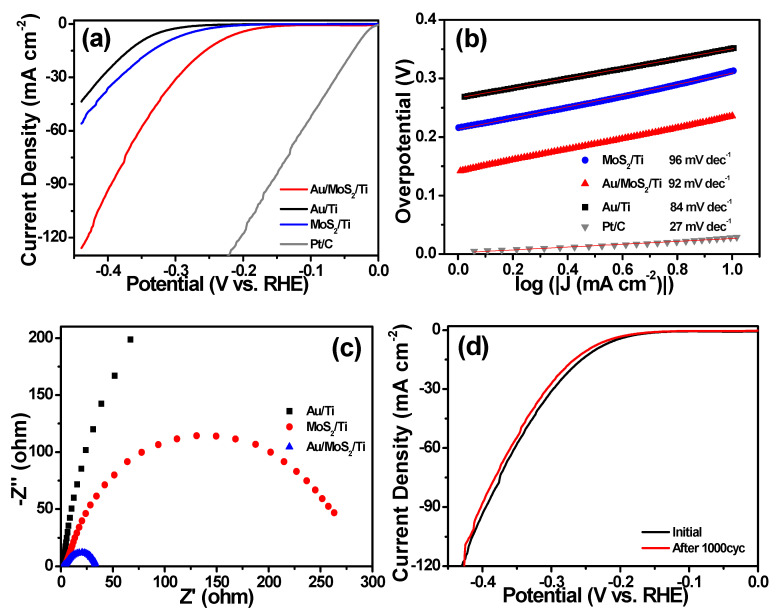
(**a**) Polarization curves and (**b**) Tafel curves of Au, MoS_2_, Au/MoS_2_ and commercial Pt/C electrocatalysts; (**c**) Nyquist plots of Au, MoS_2_ and Au/MoS_2_; (**d**) the LSV curves of Au/MoS_2_ before and after 1000 cycles.

**Figure 5 molecules-26-07503-f005:**
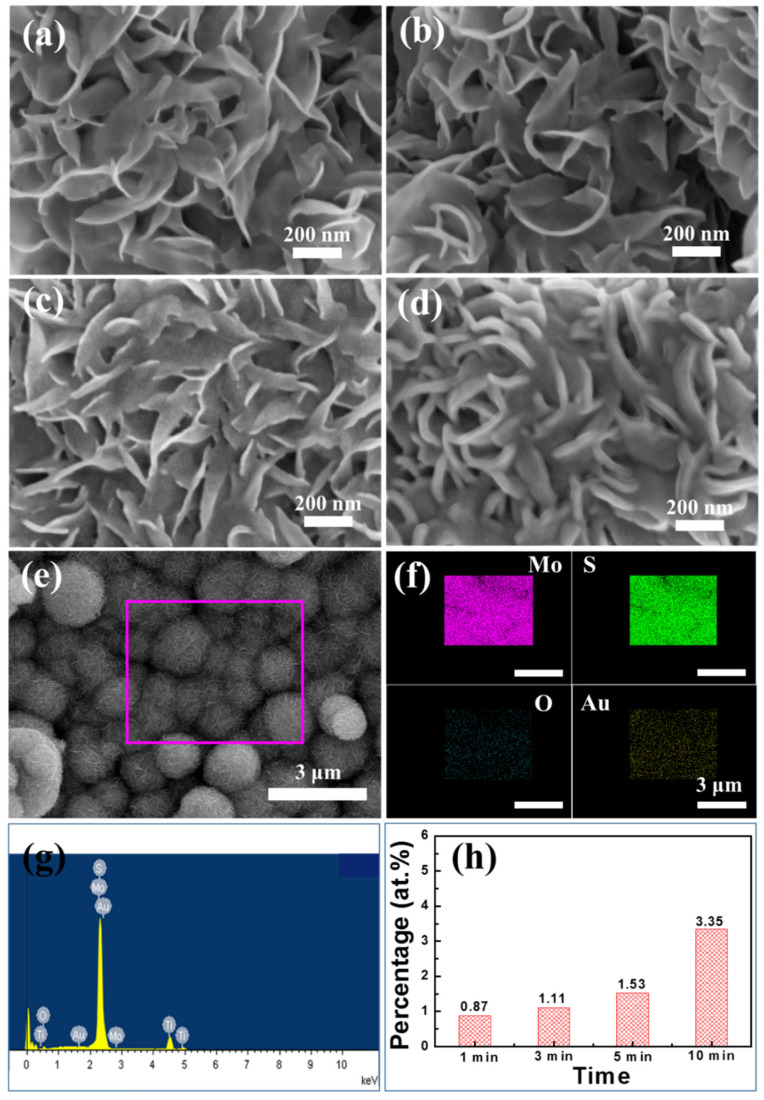
FESEM images of Au/MoS_2_ with different content of Au on the surface of MoS_2_ nanosheets: (**a**) Au-1/MoS_2_, (**b**) Au-3/MoS_2_, (**c**) Au-5/MoS_2_ and (**d**) Au-10/MoS_2_; (**e**,**f**) FESEM-mapping element distribution and (**g**) FESEM-EDS element content of Au-5/MoS_2_ and (**h**) Au content on Au/MoS_2_.

**Figure 6 molecules-26-07503-f006:**
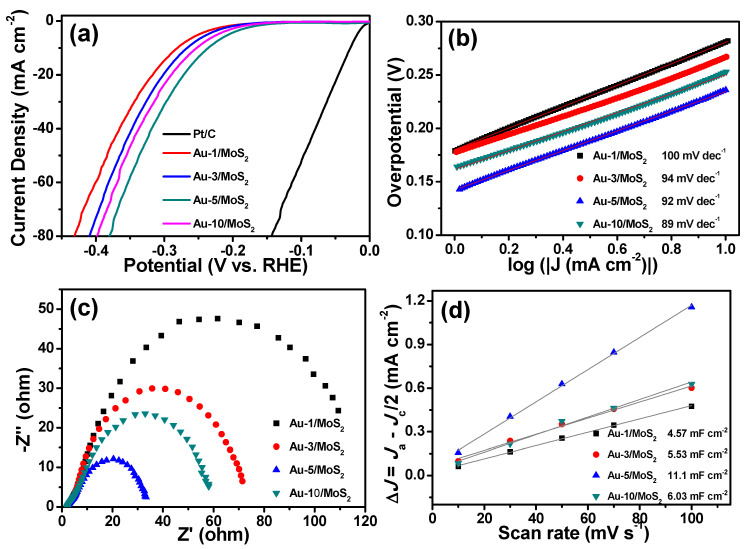
Polarization curves (**a**), Tafel curves (**b**), Nyquist plots (**c**) and the double-layer capacitance (C_dl_) (**d**) of Au/MoS_2_ with different content of Au on the surface of MoS_2_ nanosheets.

**Figure 7 molecules-26-07503-f007:**
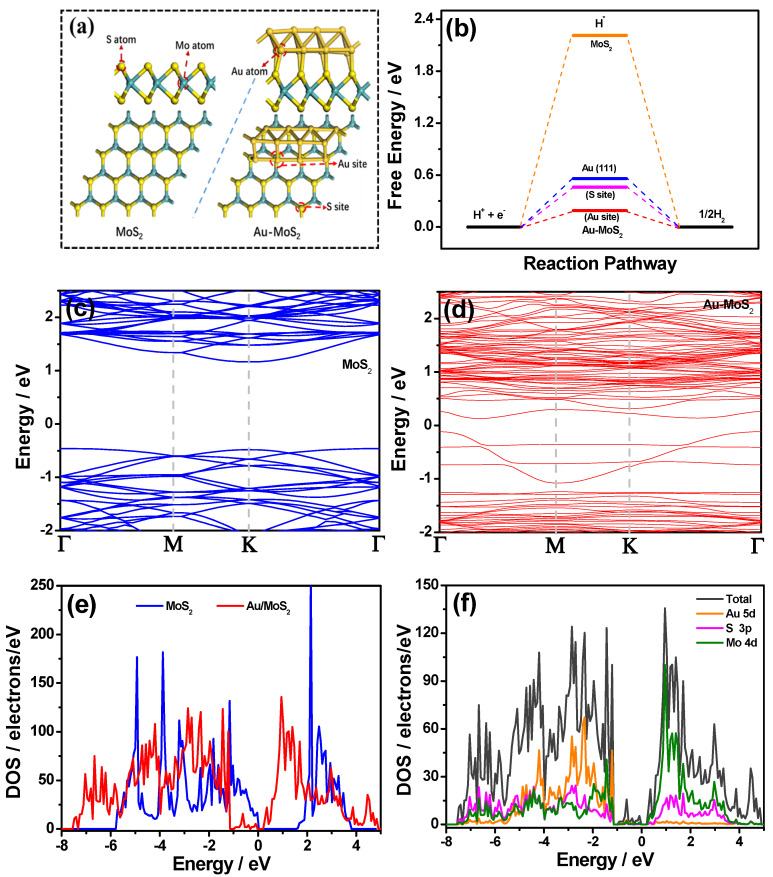
(**a**) The model of MoS_2_ and Au/MoS_2_, (**b**) the calculated free energy diagram for hydrogen evolution of Au-MoS_2_ catalysts; (**c**,**d**) band structure of MoS_2_ and Au/MoS_2_, (**e**) TDOS of MoS_2_ and Au/MoS_2_ and (**f**) PDOS of Au/MoS_2_.

## Data Availability

The data presented in this study are available in article.

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
