# Peer review of "Gold Nanoclusters Grown on MoS2 Nanosheets by Pulsed Laser Deposition: An Enhanced Hydrogen Evolution Reaction"

_molecules, 2021, doi:10.3390/molecules26247503_

Round 1

Reviewer 1 Report

Suggest following changes to the author before it can get published.

  1. For all figures 1, 2, 3, 4, 5, 6, 7: Please put (a), (b), (c) and so on before the description of the figure. For example: Figure 1: (a) SEM images of the Au/Ti, (b) MoS2/Ti (b) and (c) Au/MoS2/Ti......so on.... Current way is confusing to follow as a reader and not the general way used in the research writing.
  2. Page 2, Line 72, Section 2.2: please mention exact temperature instead of just saying room temperature
  3. The level of English of this manuscript is excellent and okay for publication after above minor changes are made.
  4. The introduction provided by the authors is appropriate and conclusions are supported by the data.

Reviewer 2 Report

The subject of the manuscript “Gold Nanoclusters Grown on MoS2 Nanosheets by Pulsed Laser Deposition: an Enhanced Hydrogen Evolution Reaction” by Yuting Jing et al. is focused on the fabrication by pulsed laser deposition (PLD) of ligand-free Au nanoparticles onto 2H MoS2 surface to form composite (Au/MoS2) for the improvement of electrocatalytic hydrogen evolution reaction (HER) activity.   

The idea of the manuscript is interesting. It can be accepted for publication after the authors will address all the raised queries (in the order they appear in the manuscript):  

  1. “1 min, 3 min, 5 min and 10 min” (page 2, line 74) – for a better/easier identification of these samples, the authors should give them some codes, starting from A, B, C and D, or Au1, Au3, Au5 and Au10.
  2. What were the values of the laser energy and spot used by the authors in their experiments? Keeping this in mind, the reader can easily obtain the corresponding value of the laser fluence. In addition, what was the distance between the Au target and the substrates? What was the value of the pressure used during the depositions? These info are important and should appear in the main text.
  3. How many of “identical” samples were in fact prepared by the authors?
  4. “This elaborately interfacial……between deposition pulses (page 3, lines 133 to 137) – this extremely long phrase is quite hard to be followed. The authors should therefore split it into at least two smaller ones.
  5. “SEM images” (page 4, line 12 – legend of Figure 1) should read as “FESEM images”.
  6. The Au/MoS2 sample (with different Au deposition times of 1, 3, 5 or 10 min) which was characterized by the authors should be clearly indicated in the legends of Figures 2, 3 and 4. Moreover, the authors should also explain the reason why they have chosen to investigate a specific sample only.
  7. “The exchange current density (j0) is an inherent feature of HER catalyst and is usually obtained by an extrapolation method.” (page 6, lines 187 to 189) – at least one reference should be included here.

Some minor recommendations follow:

- “reactant to prepared to MoS2” (page 2, line 65) should be rephrased.

- “using a molecular turbo” (page 2, line 71) should read as “using a turbo molecular pump.”.

- “which corresponds” (page 5, line 150) should read “which correspond”.

- “with the deposition time increasing.” (page 8, lines 212 to 213) should read as “with the increase of the deposition time.”

- “the MoS2 increasing” (page 8, line 230) should read as “the MoS2 increased”.
